# The Evolution of Hematopoietic Stem Cell Transplantation to Overcome Access Disparities: The Role of NMDP

**DOI:** 10.3390/cells13110933

**Published:** 2024-05-29

**Authors:** Steven M. Devine

**Affiliations:** 1NMDP, Minneapolis, MN 55401, USA; sdevine2@nmdp.org; 2CIBMTR (Center for International Blood and Marrow Transplant Research), NMDP, Minneapolis, MN 55401, USA

**Keywords:** mismatched unrelated donors, HLA, NMDP, hematopoietic stem cell transplantation (HSCT), CIBMTR, outcomes, graft-versus-host disease, donor for all

## Abstract

NMDP recognizes that despite advances in hematopoietic stem cell transplantation (HSCT) and other cell therapies, not all patients have equitable access to treatment, and disparities in outcomes remain. This paper explores the recent work of NMDP to accelerate progress and expand access to more patients through transformative clinical research, particularly in the use of mismatched unrelated donors for HSCT.

## 1. Introduction

The treatment paradigm for hematopoietic stem cell transplantation (HSCT) has changed dramatically in the last decade. Today, the use of mismatched donors and cord blood units, along with the advent of new cell and gene therapies, means treatments are available for more patients than ever before [1].

However, despite advances in HSCT and other cell therapies, NMDP recognizes that gaps in equitable access to treatment and treatment outcomes remain. NMDP has therefore developed strategies designed to achieve the vision that every patient can receive their life-saving cell therapy with the best possible outcome. One strategy involves growing and diversifying its donor registry so that more patients can find matched donors, regardless of their ancestry. However, this will take time. The other addresses short-term gaps in access to HSCT by improving the outcomes in recipients of mismatched unrelated donor (MMUD) grafts. We will describe these efforts and how the focus on both matched and MMUD transplants have evolved over the years.

## 2. The History of NMDP

NMDP exists today because the family of 10-year-old Laura Graves turned to alternative treatment options after she relapsed following chemotherapy. In 1979, Laura received the first-ever unrelated donor transplant to treat leukemia. Until that time, doctors only used family members as donors for HSCT to treat leukemia. The national registry of potential donors was created thanks to the advocacy of Laura’s father, Robert, as well as other patients’ families, physicians, congressional support, and funding from the U.S. Navy [2].

Within one year, 10,000 people stepped forward to join the national registry of potential donors. Known as the National Marrow Donor Program at the time, NMDP began matching searching patients with unrelated donors with assistance from a computer program the University of Minnesota created. In 1987, the organization facilitated its first transplant when an unrelated donor donated marrow to a 6-year-old girl. In 1994, NMDP facilitated its first peripheral blood stem cell (PBSC) collection for unrelated HSCT. Four years later, it launched its umbilical cord blood program. By 24 April 2005, 21,245 patients had an NMDP-facilitated transplant [2]. From there, HSCT use grew rapidly and, as of 30 September 2023, NMDP has facilitated more than 130,000 transplants since 1997. Today, there are more than 9 million potential donors and 270,000 cord blood units on the NMDP Registry in the U.S. and nearly 150,000 potential donors on the NMDP Registry in Mexico [3]. Every search through NMDP provides patients with access to more than 41 million potential donors and 800,000 cord blood units through the World Marrow Donor Association (WMDA), which lists donors and cord blood units from partner registries around the world. Currently, 56% of NMDP-facilitated transplants involve either an international donor or recipient [3,4].

While NMDP started as a national donor registry, it evolved into much more. NMDP and the Medical College of Wisconsin created CIBMTR (Center for International Blood and Marrow Transplant Research) in 2004 through a joint affiliation agreement to further support groundbreaking HSCT research that would increase post-HSCT survival and decrease complications [5]. In 2016, NMDP extended its services and expertise to cell and gene therapy organizations developing and delivering next-generation therapies to help more patients [6].

In 2024, the organization took another step forward by rebranding from the National Marrow Donor Program and Be The Match to NMDP [7]. Over the past 35 years, what started as a national registry of unrelated donors has grown into a global organization with a continued focus on accelerating progress through observational and interventional research and expanding access to cell therapies through direct patient support and governmental advocacy, so all patients thrive.

## 3. Mismatched Unrelated Donor Research to Improve Access and Outcomes

Leading transformative research to overcome HLA barriers to HSCT for patients with hematologic malignancies and disorders is a major focus for NMDP. Historically, outcomes for recipients of 8/8 HLA-matched unrelated donor (MUD) transplants were superior to outcomes for those who received MMUD transplants [8]. However, the likelihood a patient will have an 8/8 HLA-matched available donor differs greatly depending on the patient’s race and ethnicity [9]. That is despite access to more than 41 million potential adult donors and 800,000 umbilical cord blood units on searches of the NMDP Registry and other registries around the world [4]. For example, patients who are Black and African American have a 29% chance of finding an 8/8 HLA-matched donor on the NMDP Registry; those who are Hispanic have a 48% chance; and those who are non-Hispanic White have a 79% chance [9,10].

The NMDP Registry is the most diverse listing of potential donors in the world, and NMDP has continually focused on adding more young potential donors to the registry with diverse races and ethnicities [9,10,11]. However, recruitment alone cannot solve the issue. A 2021 NMDP analysis of potential Black and African American donors between the ages of 18 and 35 revealed a striking finding. Even if every eligible Black or African American donor (11.3 million) joined the NMDP Registry and was available when called to donate, the match rate only improved from the current 29% to 63.7%, a nearly 40% gap [12]. In addition, according to internal NMDP data, as the U.S. population continues to diversify, the number of HLA types increases along with the complexity of HLA matching.

NMDP recognizes that, while registry recruitment alone cannot remove HLA-related access disparities to allogeneic HSCT, science can. An NMDP analysis of the NMDP Registry found the use of mismatched donors down to a 5/8 HLA mismatch level can eliminate the availability gap and expand donor choice. The analysis showed that a 7/8 match alone significantly increased the likelihood of a patient finding their optimal donor, with match rates ranging from 84% for Black and African American patients to 99% for non-Hispanic White patients. The potential donor pool increased by approximately 20 times at each additional mismatch level. An HLA mismatch level down to 5/8 raised the match likelihood to nearly 100% regardless of donor age or the patient’s race/ethnic background (Figure 1) [10].

NMDP is investing in research efforts and initiatives—broadly known as Donor for All—to establish increased access to HSCT through a series of protocols aimed at improving outcomes in recipients of MMUDs in the United States and internationally while providing outcomes comparable to those with 8/8 matched donors.

Traditionally, MMUD use was limited because relative to the use of a MUD, MMUD HSCT using calcineurin inhibitor-based graft-versus-host disease (GVHD) prophylaxis was associated with increased risk for GVHD, transplant-related mortality (TRM), and diminished overall survival (OS) [8]. The use of post-transplant cyclophosphamide (PTCy) GVHD prophylaxis, abatacept, and other novel approaches have changed those outcomes [13,14,15,16,17,18,19]. PTCy has become a standard of care that was first shown to be effective in haploidentical-related HSCT [13]. That success led NMDP to sponsor a series of clinical trials studying the use of PTCy in MMUD HSCT.

The first clinical trial, known as the 15-Mismatched Unrelated Donor (15-MMUD) trial (NCT02793544), was a prospective, multi-center, Phase II study of adults who received bone marrow (BM) matched at 4/8 to 7/8 HLA alleles and PTCy, sirolimus, and mycophenolate mofetil (MMF) GVHD prophylaxis. Approximately 48% of patients enrolled in the trial were of a race/ethnicity other than non-Hispanic White. Patients in the clinical trial had an OS of 76% at 1 year, which exceeded the primary endpoint of greater than 65% OS [15]. A 3-year analysis demonstrated outcomes that remained very good. Patients who received reduced-intensity conditioning (RIC) in particular had very good outcomes with 70% OS. When comparing HLA match grades of 7/8 versus 4–6/8, OS did not differ significantly (63% for 7/8 strata and 71% in the 4–6/8 strata) [16].

Building on the success of 15-MMUD, NMDP sponsored the ACCESS clinical trial (NCT04904588), a prospective, multi-center, Phase II study of MMUDs using peripheral blood stem cells (PBSC) for adult patients (two strata) and bone marrow for pediatric patients (one stratum). The switch to PBSC as the graft source in adult recipients was important as PBSCs are safer for donors, less complicated to supply due to increased capacity relative to BM, and more predictable in terms of product quality. Patients receive PTCy, tacrolimus, and MMF GVHD prophylaxis and are matched from 4/8 to 7/8 HLA alleles. The ACCESS trial completed enrollment for the two adult strata ahead of schedule. The adult RIC stratum accrued so briskly that it was reopened at the request of investigators to include 100 additional patients for a total of 170 patients, which will allow for more granular data on outcomes for patients who receive 7/8 MMUD versus 4/8 to 6/8 MMUD HSCT. Of note, more than 50% of the patients enrolled are people of color. Results from the first 70 patients transplanted on the RIC arm should be available shortly. The pediatric stratum continues to enroll patients [20].

The NMDP-sponsored OPTIMIZE clinical trial (NCT06001385) follows in the footsteps of the ACCESS trial and enrolled its first patient in January 2024. The plan is to enroll approximately 150 patients at up to 30 transplant centers across the United States with study completion estimated in 2026. The OPTIMIZE trial aims to understand if using a reduced PTCy dose after a patient receives PBSCs from a MMUD can lower the occurrence of infections in the first 100 days after HSCT, while maintaining the same level of GVHD protection as the standard PTCy dose [21]. The standard PTCy dose has a significant risk for severe side effects including bacterial and viral infections [22]. NMDP believes the OPTIMIZE trial has the potential to transform care for patients with hematologic malignancies by

Improving patient survival and quality of life by decreasing acute and long-term toxicities associated with standard dose PTCy;Decreasing relapse by enabling increased use of myeloablative conditioning in the MMUD setting;Permitting future clinical trials that incorporate reduced dose PTCy in combination with other novel immune therapies to prevent GVHD and malignant disease relapse [21,23].

OPTIMIZE also lays the groundwork for future clinical trials that could apply MMUD HSCT to cure non-malignant diseases such as sickle cell disease. In the U.S., sickle cell disease primarily impacts people who are Black or African American, a population that has the lowest likelihood of having an 8/8 HLA-matched available unrelated donor [9,10,24].

Along with clinical trials in MMUD HSCT using PTCy prophylaxis, NMDP is committed to exploring differences in OS and GVHD-free, relapse-free survival (GRFS) rates between various unrelated donor sources. A 2023 observational study from CIBMTR compared OS and GRFS between patients who received 8/8 MUD allografts and PTCy prophylaxis and those who received 7/8 MMUD allografts and PTCy prophylaxis. The investigators analyzed data from the CIBMTR database of 4829 patients who underwent their first allogeneic HSCT between 2017 and 2022 to treat acute lymphoblastic leukemia (ALL), acute myeloid leukemia (AML), or myelodysplastic syndromes (MDS). The analysis showed no notable differences in OS and GRFS between 8/8 (N = 1517) and 7/8 (N = 540) recipients for up to 3 years post-HSCT. The rates of relapse and non-relapse mortality were consistent across the two groups regardless of conditioning intensity. The study did find MUD HSCT was associated with a slightly reduced risk of moderate/severe chronic GVHD compared to 7/8 MMUD [14].

NMDP internal operations data indicates that studies into MMUD HSCT are changing clinical practice with a rapid increase in MMUD use. NMDP-facilitated mismatched transplants are up 60% in the United States when comparing fiscal year (FY) 2021 (1 October 2020–30 September 2021) to FY2023 (1 October 2022–30 September 2023). Most of the growth comes from the use of 7/8 MMUDs; however, the unpublished NMDP operations data shows a 24% growth in the use of 4/8, 5/8, and 6/8 MMUDs from FY2021 through FY2023.

## 4. Research and Advocacy to Create Meaningful Change for All Patients

Research into the use of MMUDs opens the door to transplants for many more patients in need, but NMDP’s work does not stop there. The organization is committed to improving access and outcomes for all patients, including those with 8/8 MUDs and those who need cell therapy other than HSCT.

An observational study conducted by CIBMTR researchers was presented in abstract form at the 2023 American Society of Hematology annual meeting and assessed the impact of donor characteristics on patients’ OS and event-free survival (EFS) for patients who receive an 8/8 MUD transplant to treat hematologic malignancies and disorders. The researchers used an advanced machine learning technique to better understand how 8/8 matched unrelated donors should be prioritized based on additional donor characteristics such as age, gender, prior pregnancy, cytomegalovirus (CMV) status, or extended HLA matching (HLA-DQB1 and DPB1) [25]. Previous research defined the best donor as the youngest donor available on the search, but the study only looked at OS up to 1-year post-HSCT [26]. This study looked at OS and EFS for up to 3 years post-HSCT. Understanding the factors that impact OS and EFS could expand donor choice beyond the youngest donor.

This study included more than 11,800 first 8/8 MUD transplants performed in the U.S. and reported to the CIBMTR Research Database from 2016 to 2019, as well as a subset of 699 patients who had detailed donor search data available in the NMDP Search Archive. This study revealed that only donor age and donor gender had clinically significant impacts on outcomes. Donor age was more important for OS than EFS. An 18-year-old donor was associated with a clinically meaningful increase in OS (greater than 1%) at 3 years when compared to those 34 years old and older. However, when compared to donors aged 22 to 30, an 18-year-old donor had limited impact on overall survival (less than 1% improvement at 3 years). The ability to select unrelated donors between the ages of 18 and 30 with negligible impact on OS should increase choice for transplant centers and negate some of the recent concerns with donor availability.

Donor gender had a limited impact on OS but was important for EFS. Male donors were preferred when the age difference between a male and female donor was minimal. An optimization strategy balancing OS and EFS confirmed the potential for nuanced donor selection that prioritizes OS but considers EFS when OS differences are marginal. With results demonstrating minimal impact on patient outcomes among donors between the ages of 18 and 30, it allows more flexibility in donor selection and increases the likelihood a patient will have an available optimal donor [25].

When a patient has an optimal match on the registry, NMDP is committed to eliminating obstacles that could prevent the donor from donating. For example, currently, about half of potential donors are unable to donate when matched with a patient. One of the primary reasons they decline is the risk of losing their job. Donors need up to 40 nonconsecutive hours to complete their donation from the start of the donor screening process through a short recovery after donation. One of NMDP’s top federal legislative priorities is asking Congress to support the Life Saving Leave Act (H.R. 3024). Reps. Dean Phillips (D-Minn.) and Brian Fitzpatrick (R-Pa.) introduced the legislation in the U.S. House last year and Sens. Bob Casey (D-Pa.) and Bill Cassidy (R-La.) recently filed a U.S. Senate companion bill. H.R. 3024 would amend the Family and Medical Leave Act (FMLA) to allow donors to have up to 40 non-consecutive, unpaid hours of leave without risk to their jobs [27,28].

NMDP also understands that HSCT may not be the best treatment for every patient who needs a life-saving cure and formed NMDP BioTherapies in 2016. NMDP BioTherapies builds on the infrastructure and expertise NMDP developed matching patients and donors and coordinating the time-sensitive collection and delivery of donor cells to patients around the world. NMDP BioTherapies partners with more than 40 cell and gene therapy developers to advance clinical trials and support commercial product launches and ongoing delivery. CIBMTR also collects data on the long-term safety and efficacy of cell therapies.

Finally, NMDP recognizes that to create meaningful change in improving patient access and outcomes inequities in transplant and cell therapy, it takes active, sustained engagement from across the transplant and cell therapy ecosystem. NMDP and the American Society for Transplantation and Cellular Therapy (ASTCT) launched the ASTCT-NMDP ACCESS Initiative in 2022 to bring the HSCT and cell therapy communities together to identify and implement practice and policy changes to reduce treatment barriers and improve patient outcomes. The initiative is focused on three key areas to start:Increasing awareness among hematologists oncologists and patients;Reducing poverty-related barriers through patient, center, and policy initiatives;Improving access and outcomes regardless of race and ethnicity.

NMDP and ASTCT recognize the problems are multi-faceted and entrenched in the U.S. healthcare system and have committed to a multi-year effort to work with stakeholders throughout the HSCT and cell therapy ecosystem to overcome the challenge [29].

## 5. Future Direction

Transformative change is already underway with the Donor for All initiative and continued research into MMUD HSCT. To continue accelerating progress and expanding access, transplant centers and international registries should recognize that MMUD HSCT is proving to be safe and effective and consider adopting its use, along with haploidentical donor and umbilical cord blood donor grafts. All are viable alternative donor sources. Transplant centers are encouraged to enroll patients in clinical trials for HSCT and other appropriate cell therapies to keep the momentum moving forward. NMDP will commit to continuing its work to remove barriers for donors and ensure patients have access to their optimal donor.

NMDP got its start because of the perseverance of the Graves family who wanted to provide hope to other patients and families. It took a community of patients’ families, physicians, Congress, and the U.S. Navy for their vision to become a reality. That same perseverance drives NMDP today. It will take the entire HSCT and cell therapy community of transplant centers, cell and gene therapy developers, researchers, donors, patients and their families, and more working in partnership with NMDP and CIBMTR to make the vision of creating a world where every patient can receive their life-saving cell therapy a reality.

## Figures and Tables

**Figure 1 cells-13-00933-f001:**
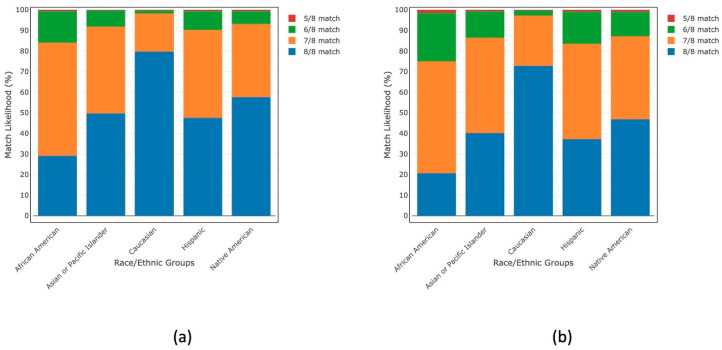
An NMDP analysis of the NMDP Registry shows match likelihood for five broad racial/ethnic groups from 5/8 to 8/8 HLA match levels for (**a**) donors for all ages and (**b**) donors age 35 or younger. The match likelihood was greater than 99% at a 5/8 or higher match level regardless of donor age or patient race/ethnicity.

## Data Availability

Not applicable.

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
