# Peer review of "The Evolution of Hematopoietic Stem Cell Transplantation to Overcome Access Disparities: The Role of NMDP"

_cells, 2024, doi:10.3390/cells13110933_

Round 1

Reviewer 1 Report

Comments and Suggestions for Authors

It shows that NMDP is committed to finding an available donor for all patients in need of transplantation, developing a novel GVHD prophylaxis using post-transplant cyclophosphamide for HLA mismatched donors, and developing legislation to allow donors to take paid leave. I think this is a valuable paper for medical professionals involved in transplantation around the world. I would like to see a little more explanation on the following points.

I am not sure how to interpret Figure 1.

What do the size and number of circles mean?

Author Response

Response to reviewer 1:

I am not sure how to interpret Figure 1.

What do the size and number of circles mean?

We will submit a higher quality figure with better figure legend

5/13/24 - Decision made to remove Figure and corresponding language.

Reviewer 2 Report

Comments and Suggestions for Authors

This is an excellent overview on the history, current activities and future directions of NMDP, with a focus on efforts to provide access to cellular therapy to all patients concerned, regardless of race and ethnicity. One major focus is the enormously increasing chance for a suitable stem cell donor by including HLA-mismatched registry donors, which is enabled by modern transplant platforms, particularly post-transplant cyclophosphamide (PTCy). 

Author Response

Thank you for this review. We hope it is helpful for the readers.

Reviewer 3 Report

Comments and Suggestions for Authors

The paper by S. Devine is very well written and provides a very interesting overview of the activities of the National Marrow Donor Program, past, present and future in regards to effort to improve accessibility of stem cell transplant to all, more precisely on improving matching for non Caucasian patients. I have no major changes recommended, just some minor edits.

Author should define the abbreviation NMDP in the abstract as many readers won’t be familiar with it, and would like to know its meaning right away.

Figure 1 is good, but the font used around it makes Fig 1 almost impossible to appreciate and understand. Written words on the left hand side are all too small and need resizing, and author should ensure that its readable at magnification 100. Same for text under Fig 1. Using Bold font can also help.

Font in fig 2 could be also increase by 1 pt or placed in bold .

Since many abbreviations are used, I wonder if the journal would allow the author to insert a list of commonly used abbreviations to help readers.

Author Response

Please see point by point response to reviewers comments:

Author should define the abbreviation NMDP in the abstract as many readers won’t be familiar with it, and would like to know its meaning right away.

NMDP has changed its brand and is now just NMDP. It formerly stood for National Marrow Donor Program

Figure 1 is good, but the font used around it makes Fig 1 almost impossible to appreciate and understand. Written words on the left hand side are all too small and need resizing, and author should ensure that its readable at magnification 100. Same for text under Fig 1. Using Bold font can also help.

We agree and will provide a higher quality figure

5/13/24 - Decision made to remove Figure and corresponding language.

Font in fig 2 could be also increase by 1 pt or placed in bold .

We agree and will provide a higher quality figure

Since many abbreviations are used, I wonder if the journal would allow the author to insert a list of commonly used abbreviations to help readers

Let us know response to this. We employed commonly used abbreviations

Reviewer 4 Report

Comments and Suggestions for Authors

The manuscript can be considered for publication.

The history and fondamental role of NMDP in donor recruitment are adequately described. In particular, the Author demonstrated that NMDP is a virtuous model for all the National Registries, as it supports studies which contribute to qualifying and improving HSCT results. 

Minor suggestions:

NMDP should be defined as National Marrow Donor Program (in the abstract or in line 19)

Figure 1: the Author should provide a definition of the meaning of the circles and their size

Author Response

Response to Reviewer 4

The history and fundamental role of NMDP in donor recruitment are adequately described. In particular, the Author demonstrated that NMDP is a virtuous model for all the National Registries, as it supports studies which contribute to qualifying and improving HSCT results. 

Minor suggestions:

NMDP should be defined as National Marrow Donor Program (in the abstract or in line 19)

NMDP is now just NMDP, formerly National Marrow Donor Program.

Figure 1: the Author should provide a definition of the meaning of the circles and their size.

We agree and will provide a better figure and legend

5/13/24 - Decision made to remove Figure and corresponding language.